# PGAM5 promotes lasting FoxO activation after developmental mitochondrial stress and extends lifespan in *Drosophila*

Martin Borch Jensen[1], Yanyan Qi[1], Rebeccah Riley[1], Liya Rabkina[1], Heinrich Jasper[1,2]*

[1]Buck Institute for Research on Aging, Novato, United States; [2]Immunology Discovery, Genentech, South San Francisco, United States

**Abstract** The mitochondrial unfolded protein response (UPRmt) has been associated with long lifespan across metazoans. In *Caenorhabditis elegans*, mild developmental mitochondrial stress activates UPRmt reporters and extends lifespan. We show that similar developmental stress is necessary and sufficient to extend *Drosophila* lifespan, and identify Phosphoglycerate Mutase 5 (PGAM5) as a mediator of this response. Developmental mitochondrial stress leads to activation of FoxO, via Apoptosis Signal-regulating Kinase 1 (ASK1) and Jun-N-terminal Kinase (JNK). This activation persists into adulthood and induces a select set of chaperones, many of which have been implicated in lifespan extension in flies. Persistent FoxO activation can be reversed by a high-protein diet in adulthood, through mTORC1 and GCN-2 activity. Accordingly, the observed lifespan extension is prevented on a high-protein diet and in FoxO-null flies. The diet-sensitivity of this pathway has important implications for interventions that seek to engage the UPRmt to improve metabolic health and longevity.

DOI: https://doi.org/10.7554/eLife.26952.001

*For correspondence:
hjasper@buckinstitute.org

Competing interests: The authors declare that no competing interests exist.

## Introduction

A wide range of studies in genetically accessible model systems have led to the realization that aging is a malleable process, responsive to both genetic and pharmacological interventions. An integrated view of the aging process has emerged from these efforts, spurred by the identification of a select group of biological processes and pathways that drive, influence, and regulate the physical decline characteristic of the aging process (*Kennedy et al., 2014*; *López-Otín et al., 2013*). Many of these pathways involve mitochondria: either through their role in metabolism (*López-Otín et al., 2016*), as a source of reactive oxygen species (*Balaban et al., 2005*), or as signaling hubs (*Chandel, 2015*).

In recent years, the mitochondrial unfolded protein response (UPRmt) has emerged as a unifying mechanism for several of these pathways (*Jensen and Jasper, 2014*). As the name implies, the UPRmt is a conserved cellular mechanism that serves to restore proteostasis in mitochondria. The response was first described in mammalian cells more than a decade ago (*Martinus et al., 1996*; *Zhao et al., 2002*) and has since been studied primarily in *Caenorhabditis elegans*. In worms, strong evidence suggests that the UPRmt is involved in delaying aging and promoting adult lifespan (*Baker et al., 2012*; *Durieux et al., 2011*; *Houtkooper et al., 2013*; *Merkwirth et al., 2016*; *Mouchiroud et al., 2013*; *Tian et al., 2016*; *Yang and Hekimi, 2010*). This work has revealed the UPRmt to be reminiscent of, but distinct from, the cytoplasmic (heat shock) and endoplasmic reticulum unfolded protein responses (*Baker et al., 2012*; *Haynes et al., 2007*; *2010*). Through the UPRmt, mitochondrial stress induces a nuclear transcriptional response that promotes the expression of a group of mitochondrial chaperones and proteases (*Aldridge et al., 2007*; *Yoneda et al., 2004*).

The primary transcription factor in *C. elegans* is ATFS-1, which is regulated by membrane potential-dependent import into and degradation in mitochondria (*Nargund et al., 2012*). Mitochondrial stress blocks this import and ATFS-1 instead moves to the nucleus, where it interacts with DVE-1 and UBL-5 to turn on the transcriptional response (*Benedetti et al., 2006*; *Haynes et al., 2007*). In addition to proteostatic elements, this response includes a metabolic reconfiguration to increase glycolytic capacity while restoring oxidative phosphorylation (*Nargund et al., 2015*). Meanwhile, a separate branch of the UPR^mt mediates a general downregulation of translation, through GCN-2 and eIF2α (*Baker et al., 2012*).

Evidence for the evolutionary conservation of the UPR^mt has emerged in recent years. In mice, perturbation of mitochondrial translation has been implicated in long lifespan (*Houtkooper et al., 2013*), and recent findings have revealed the conservation of the ATFS-1 regulated transcriptional response (mediated by ATF5 in mice (*Fiorese et al., 2016*)). In *Drosophila*, a UPR^mt-like response was first described in a paradigm where a misfolding ornithine transcarbamylase (ΔOTC) was overexpressed, resulting in upregulation of mitochondrial chaperones and induction of mitophagy (*Pimenta de Castro et al., 2012*). Furthermore, knocking down electron transport chain (ETC) components has been shown to extend lifespan and to induce the UPR^mt (*Copeland et al., 2009*; *Owusu-Ansah et al., 2013*). However, the signaling pathway mediating UPR^mt activation in *Drosophila* remains to be clarified. ETC knockdown results in induction of the insulin signaling inhibitor ImpL2, and promotes the expression of target genes of the insulin-regulated transcription factor Forkhead Box O (FoxO) (*Owusu-Ansah et al., 2013*). FoxO activation is a well-established lifespan-extending condition, yet its involvement and specific regulation in this context remain to be established (*Kappeler et al., 2008*; *Kenyon et al., 1993*; *Kimura et al., 1997*; *Selman et al., 2008*; *Tatar et al., 2001*).

Activation of the UPR^mt has also been implicated in the lifespan-extending effects of various drugs, including resveratrol (*Houtkooper et al., 2013*), rapamycin (*Houtkooper et al., 2013*) and NAD precursors (*Mouchiroud et al., 2013*). Nevertheless, it is not clear which cellular consequences of the UPR^mt contribute to organismal health, nor whether resistance to a particular type of stress underlies lifespan extension by the UPR^mt. Indeed, the generic view that stimulating this response invariably leads to longer life has been called into question (*Bennett et al., 2014*). Understanding how conserved elements of UPR^mt signaling connect to longevity pathways will be useful in resolving this contention.

An interesting observation from both fly and worm studies is that lifespan extension by the UPR^mt is contingent upon its activation during development (*Durieux et al., 2011*; *Owusu-Ansah et al., 2013*). This is consistent with earlier observations that lifespan extension by disrupting the ETC in some cases require developmental treatment (*Copeland et al., 2009*; *Dillin et al., 2002*; *Rea et al., 2007*). An important step toward understanding this lasting effect came from work in worms, in which the UPR^mt leads to changes in histone methylation and chromatin organization (*Merkwirth et al., 2016*; *Tian et al., 2016*). The H3K27 demethylases *jmjd-1.2* and −*3.1* were found to be required for UPR^mt activation and lifespan extension, while the histone methylase *met-2* is required for induction of most UPR^mt genes. This effect involves global chromatin condensation, while opening up specific sites for occupation by DVE-1 (*Tian et al., 2016*). The loci thus revealed have yet to be characterized, so the lasting cellular changes and longevity pathways influenced by these epigenetic mechanisms remain unclear.

Here, we have explored the signaling pathway regulating the transcriptional response to mitochondrial proteostatic stress in *Drosophila*, and have identified a role for persistent FoxO activation in promoting longevity after developmental mitochondrial stress. We find that Phosphoglycerate Mutase 5 (PGAM5) is required to activate a pathway that includes ASK1, JNK, Relish and FoxO and promotes protective gene expression in response to mitochondrial stress. Activation of this pathway during development leads to persistent FoxO activation and increased expression of chaperones in adult flies, and is required for longevity. We further find that lasting FoxO activation is sensitive to dietary conditions, as it can be abolished by elevated protein intake and elevated mTORC1 and GCN-2 activity. Our findings identify a new diet-sensitive pathway of lifespan regulation by mitochondrial stress. Since the identified pathway components are evolutionarily conserved, we anticipate that these results inform our understanding of similar interactions in vertebrate systems, including humans.

## Results

### Acute mitochondrial stress upregulates proteostatic, immune and stress signaling pathways through PGAM5- and JNK-dependent FoxO activation

To gain an overview of the gene expression changes induced by the UPR$^{mt}$ in *Drosophila*, we sequenced mRNA extracted from adult female thoracic tissue after 24 hr of mitochondrial stress. We induced mitochondrial stress ubiquitously, using GeneSwitch for temporal control, either by overexpressing a misfolding variant of the human mitochondrial enzyme ornithine transcarbamylase (ΔOTC) (*Pimenta de Castro et al., 2012*) or by knocking down the ETC complex I component ND75 (*Owusu-Ansah et al., 2013*). 23% of the ΔOTC-induced genes and 15% of the genes downregulated by ΔOTC expression were co-regulated by both conditions, and we propose that these genes compose a core UPR$^{mt}$ in *Drosophila* (*Figure 1a*). The main functional categories enriched among the induced genes were: several stress signaling pathways, chaperones and proteases, and the innate immune system (*Figure 1b*). Activation of immune responses is broad in this case, encompassing target genes of both the *Toll* and *Imd* pathways.

We confirmed the induction of several immune genes by quantitative real-time PCR (qPCR), and further demonstrated that transcriptional induction of these genes was specifically triggered by mitochondrial stress and not by conditions that trigger the cytoplasmic- (heat shock) or endoplasmic reticulum (loss of ER-associated degradation after knockdown of Hrd1, (*Bordallo et al., 1998*)) unfolded protein responses (*Figure 1c*). We also used qPCR to confirm that ΔOTC expression leads to induction of mitochondrial chaperones and proteases associated with the UPR$^{mt}$, under conditions used in earlier studies (*Figure 1—figure supplement 1a*). As further confirmation of immune activation, we tested the ability to fight off pathogenic bacteria introduced either orally or by abdominal pricking, following 24 hr of ΔOTC expression. Resistance to oral infection was mildly increased, while the speed by which flies succumb to humoral infection was unaffected; we speculate that this may be because the modest induction of immune genes is not sufficient to clear bacteria introduced directly to the hemolymph (*Figure 1—figure supplement 2a*). Initial experiments confirmed these phenotypes in both male and female flies (data not shown), and we opted to use females for all following experiments.

Activation of a transcriptional program that includes immune response genes by the UPR$^{mt}$ has previously been noted in worms, where both immune and proteostatic gene induction is dependent on the ATFS-1 transcription factor (*Nargund et al., 2012*; *Pellegrino et al., 2014*). Since a homologous mediator of the response to mitochondrial stress in flies had not been described, we set out to identify transcription factors that could mediate the UPR$^{mt}$ in this organism. We performed a targeted RNAi screen of transcription factors known to regulate immune and stress genes. Of the two methods initially used to induce the mitochondrial proteostatic stress, we opted to use ΔOTC for further experiments to minimize potential secondary effects of inducing ETC dysfunction. Because of its robust, specific and stable induction, we focused on the expression of the antimicrobial peptide *metchnikowin* (*Mtk*) as a reporter of mitochondrial stress. Knocking down the transcription factors *foxo* and *Relish*/NF-κB, but not *jun* or *kayak*/Fos, strongly suppressed *Mtk* induction (*Figure 1d*), consistent with upregulation of several FoxO target genes in our RNAseq data (*Figure 1a*). Because RNAseq further suggested activation of the c-Jun N-terminal kinase (JNK) pathway, and since JNK has been reported to regulate FoxO (*Wang et al., 2005*), we asked whether knocking down the *Drosophila* JNK *basket* (*bsk*) would prevent *Mtk* induction. Indeed, *Mtk* induction by mitochondrial stress was significantly impaired in *bsk* loss-of-function conditions (*Figure 1d*).

We next explored mechanisms by which mitochondrial stress could activate JNK. One of the JNK-activating kinases, Apoptosis Signaling Kinase 1 (ASK1), had been reported to interact with and be activated by the mitochondrial Serine/Threonine phosphatase Phosphoglycerate Mutase 5 (PGAM5) (*Takeda et al., 2009*). Knocking down *ASK1* and *PGAM5* blocked induction of *Mtk* after mitochondrial stress, suggesting that PGAM5, ASK1, and JNK are components of a pathway responsible for transducing mitochondrial stress to downstream transcription factors (*Figure 1d*). The lack of *Mtk* induction in *PGAM5* homozygous null mutants (*PGAM5$^{1/1}$*) experiencing mitochondrial stress confirmed this finding (*Figure 1d*). To determine whether this pathway indeed represented UPR$^{mt}$ signaling, we tested and confirmed that its disruption also blocks induction of classical UPR$^{mt}$ genes such as *hsp60* and *hsp10* (*Figure 1—figure supplement 1b–d*).

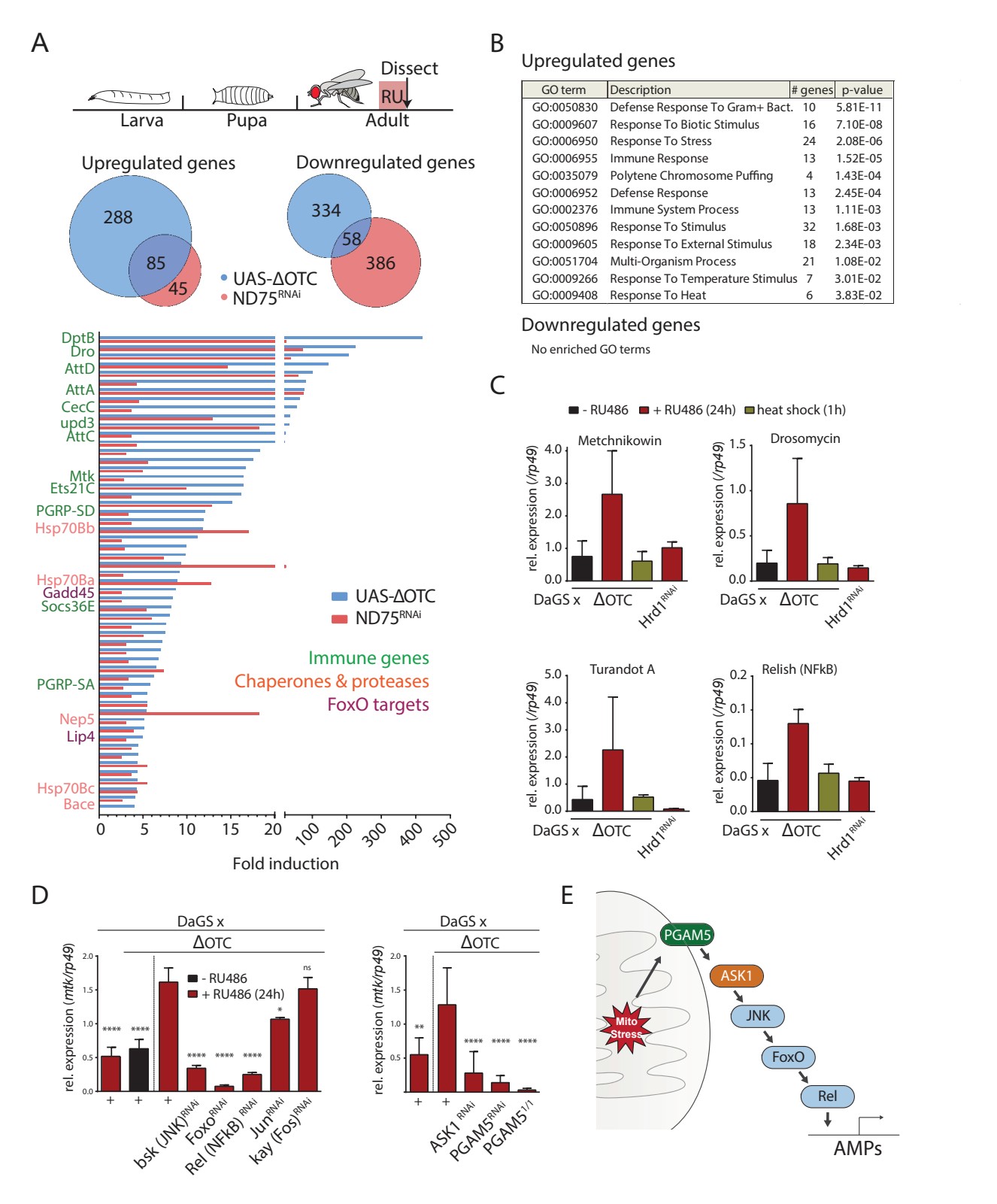

**Figure 1.** The *Drosophila* signaling pathway and transcriptional response to mitochondrial proteostatic stress. (**A**) Mitochondrial proteostatic stress was triggered by overexpression of a misfolding mitochondrial protein (ΔOTC) or knockdown of the complex I subunit ND75, as previously reported. The ubiquitous DaGS::GeneSwitch driver was activated for 24 hr at day 7 of adulthood, and female thoracic tissue collected for RNAseq analysis. Venn and bar diagrams show genes upregulated 3-fold/downregulated 2-fold with both treatments relative to controls, and with control FKPM > 10. Full raw data

*Figure 1 continued on next page*

*Figure 1 continued*

found in Supplementary File 1. (**B**) Gene Ontology analysis of the shared stress response; upregulated genes include chaperones, stress pathways, and antimicrobial peptides/innate immune genes. Redundant GO terms were trimmed using REVIGO. (**C**) Mitochondrial proteostatic stress response genes from RNAseq analysis verify by qPCR and are not activated by heat shock or induction of the ER-UPR through RNAi of Hrd1. p-values for ΔOTC ± RU486 are 0.109, 0.130, 0.255 and 0.008. (**D**) Knocking down the transcription factors FoxO or Relish, or the JNK kinase, during ΔOTC expression blocks induction of response genes. The kinase ASK1 and the mitochondrial membrane protein PGAM5 are also required for the mitochondrial proteostatic stress response. See supplementary for verification of additional UPR$^{mt}$ genes. p-values relative to ΔOTC + RU are (left to right)<0.0001,<0.0001,<0.0001,<0.0001,<0.0001, 0.0185 and 0.9888; 0.002,<0.0001,<0.0001, and <0.0001. (**E**) Proposed signaling pathway for immune activation by mitochondrial proteostatic stress. All error bars are SEM of 3+ independent experiments. ANOVA with Dunnett's multiple comparisons test; *p<0.05, **p<0.01, then each *=0.1 x.. 200 μM RU486 was used in all experiments.
DOI: https://doi.org/10.7554/eLife.26952.002

The following source data and figure supplements are available for figure 1:

**Source data 1.** The *Drosophila* signaling pathway and transcriptional response to mitochondrial proteostatic stress.
DOI: https://doi.org/10.7554/eLife.26952.006
**Figure supplement 1.** Activation of UPR$^{mt}$ markers by ΔOTC expression.
DOI: https://doi.org/10.7554/eLife.26952.003
**Figure supplement 2.** Immune activation by mitochondrial proteostatic stress.
DOI: https://doi.org/10.7554/eLife.26952.004
**Figure supplement 3.** *Drosophila* UPR$^{mt}$ signaling does not depend on reactive oxygen species (ROS).
DOI: https://doi.org/10.7554/eLife.26952.005

We then asked whether *PGAM5* or *ASK1* also play a role in canonical pathways for antimicrobial peptide (AMP) induction, and measured induction of *Mtk* following exposure to the pathogenic *Pseudomonas entomophila*. Ubiquitously knocking down either *PGAM5* or *ASK1* had no effect on this response (*Figure 1—figure supplement 2b*), suggesting that the pathway mediating AMP induction in response to mitochondrial stress is distinct from the canonical *Toll* and *Imd* pathways.

Because PGAM5 has been reported to regulate apoptosis (*Ishida et al., 2012*; *Wang et al., 2012*), we further investigated whether ΔOTC-induced stress signaling was the result of increased apoptosis. Although PGAM5 null flies showed higher baseline levels of apoptosis, in line with previous reports (*Ishida et al., 2012*), ΔOTC expression stress did not induce apoptosis in our experiments (*Figure 1—figure supplement 2c*).

Since ASK1 can be activated by redox signaling (*Saitoh et al., 1998*), we also tested whether mitochondrial reactive oxygen species (ROS) production was required for transcriptional activation of AMPs. We blocked mitochondrial ROS production from sites on complexes I and III of the ETC using previously characterized compounds (*Brand et al., 2016*; *Orr et al., 2013*) but did not see any effect on *Mtk* induction after mitochondrial stress (*Figure 1—figure supplement 3a*). Along these lines, overexpression of the antioxidant enzymes Catalase and jafrac1/Trx-1 during developmental ΔOTC expression does not affect acute or lasting FoxO activation (*Figure 1—figure supplement 3b & c*). This suggests that direct activation of ASK1 by ROS is not the basis of ΔOTC-mediated AMP induction.

Together, these data indicate that the transcriptional response to mitochondrial unfolded proteins in flies is mediated by a pathway involving PGAM5 and ASK1, as well as JNK, Relish and FoxO (*Figure 1e*).

## Mitochondrial proteostatic stress does not induce longevity via improved adult immune function, or via changes to the microbiome

In both flies and worms, the UPR$^{mt}$ has been reported to extend lifespan when activated during development (*Durieux et al., 2011*; *Owusu-Ansah et al., 2013*). To confirm this observation in our system, we used the ubiquitous RU486-inducible *daughterless* GeneSwitch (DaGS) driver to express ΔOTC throughout larval development only, and measured adult lifespan. As shown in *Figure 2a,g—*figure supplement 2, this treatment consistently extended maximum and median lifespan. Confirming observations in *C. elegans*, ΔOTC expression limited to adulthood did not lead to extended lifespan (*Figure 2b*). To determine whether the pathway we identified in the experiments described above is required for this longevity effect, we assessed lifespan in PGAM5 homozygous null flies (*Figure 2c*). The absence of PGAM5 prevented lifespan extension after developmental ΔOTC

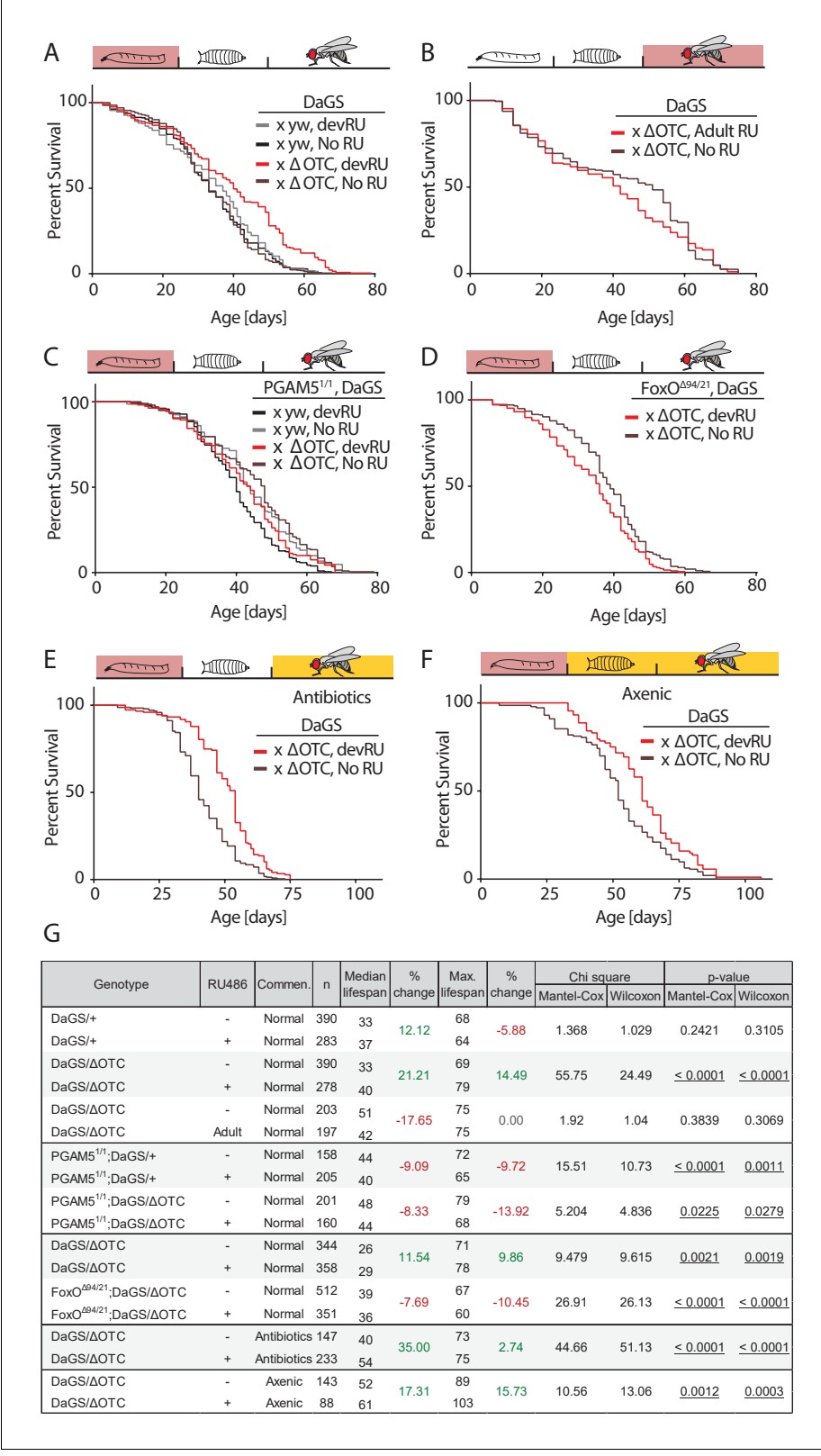

**Figure 2.** Developmental mitochondrial stress extends adult lifespan. (**A**) Expressing ΔOTC throughout larval development extends median and maximal adult lifespan by 21/14%, respectively. RU486 at 200 µM does not affect lifespan in genetic control flies. (**B**) Expressing ΔOTC throughout adulthood does not extend lifespan. (**C**) Null mutants for the mitochondrial protein PGAM5 do not live longer following developmental ΔOTC expression. (**D**) Null mutants for FoxO do not live longer following developmental ΔOTC expression. These flies also showed strongly reduced survival

*Figure 2 continued on next page*

*Figure 2 continued*

through developmental mitochondrial stress (see supplementary). (E) Flies given antibiotics throughout adulthood still show lifespan extension by developmental ΔOTC expression. (F) Flies raised in a sterile environment from egg stage still show lifespan extension by developmental ΔOTC expression. (G) Statistical analysis of lifespan experiments. Each graph contains data pooled from 2 + independent experiments. Individual experiments, mortality graphs and life tables are shown in Supplement 4 and File 2.

DOI: https://doi.org/10.7554/eLife.26952.007

The following figure supplements are available for figure 2:

**Figure supplement 1.** Survival through development stress requires UPR<sup>mt</sup> signaling.

DOI: https://doi.org/10.7554/eLife.26952.008

**Figure supplement 2.** Individual longevity curves.

DOI: https://doi.org/10.7554/eLife.26952.009

**Figure supplement 3.** Developmental ΔOTC expression does not produce long-term changes in pathogen resistance or to the microbiome.

DOI: https://doi.org/10.7554/eLife.26952.010

**Figure supplement 4.** Mortality rates for all lifespan experiments.

DOI: https://doi.org/10.7554/eLife.26952.011

expression, suggesting that the protective effects responsible for lifespan extension are downstream of PGAM5. In line with this, the adult lifespan of FoxO double heterozygous null flies was significantly reduced after developmental ΔOTC expression (*Figure 2d*). Both PGAM5 and especially FoxO nulls also showed reduced survival through larval development when subjected to mitochondrial proteostatic stress (*Figure 2—figure supplement 1*). This suggests that FoxO activity plays an integral part in promoting resilience to mitochondrial stress in these conditions.

In worms, the UPR<sup>mt</sup> also induces the expression of immune genes (*Pellegrino et al., 2014*), but it has not been explored whether UPR<sup>mt</sup>-induced longevity is a result of improved immune function. Since our data showed that the pathway responsible for activating immune-response genes during mitochondrial stress is also required for longevity, we hypothesized that the UPR<sup>mt</sup> might extend lifespan by improving the ability to fight off infections and/or by altering the microbiome. To test this hypothesis, we repeated our lifespan experiments in two conditions where flies are not exposed to microbes; in one condition, eggs were washed in bleach (HClO) to eliminate microbes, then transferred to a sterile hood and reared on autoclaved food throughout development and adult life. In the second condition, we added a cocktail of antibiotics shown to eliminate all culturable strains of the microbiome (*Li et al., 2016*) to the food of adult flies. Both conditions prevent pathogen exposure and eliminate the adult microbiome, while the first condition further prevents the formation of a larval microbiome. In both cases, we observed increased median and maximal lifespan after developmental ΔOTC expression, just as in flies reared normally (*Figure 2e–f*). Larval ΔOTC expression also did not affect pathogen resistance at day 7 of adulthood (*Figure 2—figure supplement 3a*). We further analyzed the intestinal microbiome load (as assessed by colony-forming units, CFUs) at different points of life after developmental ΔOTC expression, and found no significant difference relative to controls (*Figure 2—figure supplement 3b*). To test effects on microbiome composition, we performed 16S sequencing in young and old flies after developmental ΔOTC expression, and similarly found no major changes relative to controls (*Figure 2—figure supplement 3c*). Altogether, these data indicate that activation of the immune system following developmental ΔOTC expression does not contribute to the observed increase in adult longevity.

## Developmental mitochondrial proteostatic stress affects the metabolic state of adult flies and leads to persistent FoxO activation in the fat body

To further explore potential physiological mechanisms conferring lasting protective effects of developmental mitochondrial stress, we assessed whether metabolism in adult flies is affected (*Figure 3*, *Figure 3—figure supplement 1*). We first tested the ratio of NAD/NADH, as a measure of mitochondrial function, but did not detect changes in adults after developmental ΔOTC expression (*Figure 3a*). We also did not see changes in MitoTracker intensity in muscle tissue from adults after developmental ΔOTC expression (*Figure 3—figure supplement 1a*). However, we found that expressing ΔOTC during development leads to reduced triglyceride (TAGs) concentrations (relative to total protein) in adult flies (*Figure 3b*); this phenotype is consistent with activation of FoxO, which

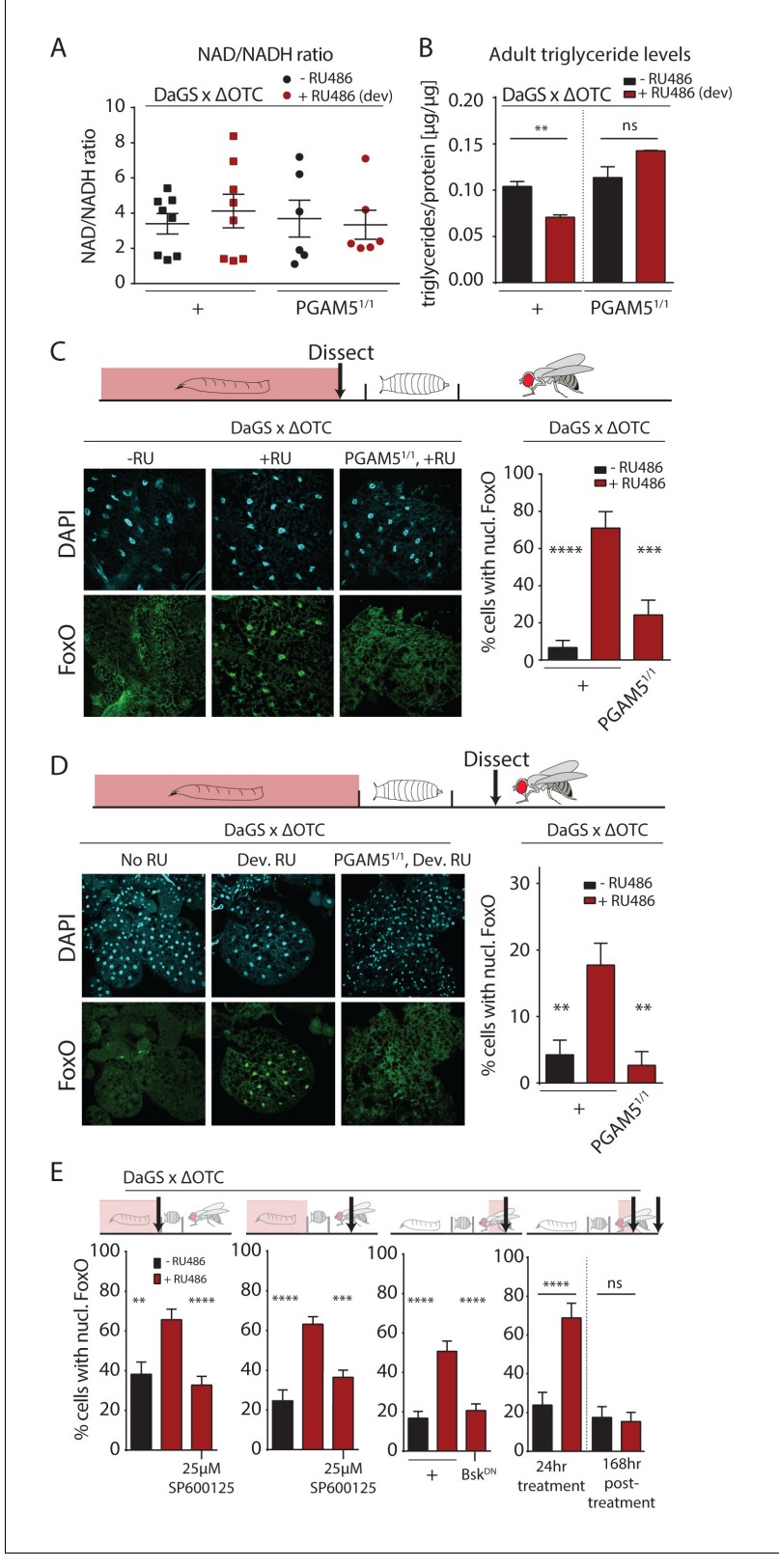

**Figure 3.** Developmental mitochondrial stress changes adult metabolism and leads to lasting activation of FoxO. (A) Developmental ΔOTC expression does not affect the ratio of NAD/NADH in adult flies. (B) Developmental ΔOTC expression lowers steady-state triglyceride levels in adult flies, consistent with fat body FoxO activation. In PGAM5 null flies this effect is not seen. p-values are 0.011 and 0.567. (C) ΔOTC expression throughout

*Figure 3 continued on next page*

*Figure 3 continued*
development leads to increased nuclear localization of FoxO in the fat body of third instar larvae. This effect is neutralized in PGAM5 null larvae. p-values are <0.0001 and 0.0001. (D) Adult fat bodies show persistent nuclear FoxO localization after developmental stress. Flies were dissected 7 days after eclosion, with no adult ΔOTC expression. p-values are 0.002 and 0.002. (E) Larval and lasting FoxO translocation by the mitochondrial stress can be blocked by simultaneously administering the JNK inhibitor SP600125 (far left and left). p-values are 0.001 and <0.0001;<0.0001 and 0.0002. Similarly, 24 hr mitochondrial stress in adult flies leads to FoxO translocation, which is blocked by simultaneous expression of dominant negative bsk/JNK (right). p-values are <0.0001 and<0.0001. In contrast to developmental treatment, FoxO activation induced by adult ΔOTC expression does not persist 1 week after treatment is stopped (far right). p-values are <0.0001 and 0.96. All error bars are SEM from two independent experiments. Student's t-test (B) or ANOVA with dunnett's multiple comparisons (A, C–E); *p<0.05, **p<0.01, then each *=0.1 x. 200 µM RU486 used in all experiments. .
DOI: https://doi.org/10.7554/eLife.26952.012
The following source data and figure supplement are available for figure 3:

**Source data 1.** Developmental mitochondrial stress changes adult metabolism and leads to lasting activation of FoxO.
DOI: https://doi.org/10.7554/eLife.26952.014
**Figure supplement 1.** Metabolic effects of developmental ΔOTC expression.
DOI: https://doi.org/10.7554/eLife.26952.013

is known to induce lipases and increase fat metabolism (*Karpac et al., 2013*). PGAM5 null mutants did not show a decrease in TAG levels, supporting the notion that this metabolic shift results from UPR^mt signaling.

The fat body is a major metabolic organ in *Drosophila*, and we have previously shown that JNK activation in the larval fat body leads to nuclear translocation and activation of FoxO (*Wang et al., 2005*). We therefore used immunohistochemistry to explore the dynamics of FoxO localization in the fat body in response to mitochondrial stress. ΔOTC expression during development strongly induced nuclear translocation of FoxO in late L3 larval fat bodies (*Figure 3c*). Again, this was not evident in PGAM5 null mutant larvae. FoxO is a known regulator of stress responses and organismal lifespan, but in order to directly promote longevity in response to developmental UPR^mt, FoxO would have to remain active during adulthood. To test this, we imaged adult fat bodies one week after eclosion and found that flies retained nuclear-localized FoxO after developmental ΔOTC expression (*Figure 3d*). We next tested whether this lasting effect on FoxO was dependent on JNK activity during development, by adding the JNK inhibitor SP600125 to the larval food. This treatment inhibited both larval and adult translocation of FoxO to the nucleus (*Figure 3e*, left). To confirm that this was due to specific inhibition of JNK/bsk, we also used genetic tools to block this pathway. Developmental expression of dominant negative bsk (bsk^DN) was lethal, but expressing bsk^DN simultaneously with ΔOTC in adult flies limited FoxO nuclear translocation (*Figure 3e*).

Since developmental activation is crucial for UPR^mt-mediated longevity ((*Durieux et al., 2011*) and *Figure 2b*), we tested whether the lasting effect on FoxO depends on the timing of ΔOTC expression. We expressed ΔOTC for 24 hr on day 3 of adulthood, which lead to acute but not lasting nuclear translocation of FoxO (*Figure 3e*, right). The absence of a persistent response to mitochondrial stress after adult treatment adults is consistent with the lack of lifespan extension in these conditions (*Figure 2b*). Our results suggest that the persistent activation of FoxO in response to the developmental activation of the UPR^mt is the mechanism by which developmental but not adult mitochondrial stress can extend lifespan.

## Heat-shock proteins are persistently upregulated in adult flies after developmental stress

To test this hypothesis, we then explored the transcriptional consequences of developmentally activated, persistent FoxO activation in adults. We performed RNA sequencing of fat bodies from both larvae acutely expressing ΔOTC, and 7-day-old adults that had experienced larval ΔOTC expression. To distinguish the specific transcriptional response responsible for longevity from more general adaptations to mitochondrial stress, we performed this experiment in both wild type and PGAM5 null mutant larvae and adult flies. We observed a large number of upregulated genes in the acutely

stressed larvae (*Figure 4a*; using a minimum baseline FPKM value of 10 and a cutoff of 3-fold induction relative to controls). These genes were predominantly involved in energy metabolism, and showed considerable overlap between the wild type and PGAM5 null mutant data sets. Only a small number of these (predominantly heat-shock and stress response genes) remained upregulated in adult flies. Notably, none of the genes induced persistently in wild-type flies remained induced in PGAM5 nulls (*Figure 4b,c*). These genes include twelve chaperones, some of which are FoxO targets (*Donovan and Marr, 2016*; *Wang et al., 2003*) and have previously been shown to extend lifespan when upregulated (*Liao et al., 2008*; *Morrow et al., 2004*; *Tatar et al., 1997*; *Wang et al., 2003*; *Zhao et al., 2005*). The only non-chaperone-encoding genes in this list (Tsp42E and Obp99) are also regulated by FoxO and are involved in lifespan regulation (*Alic et al., 2014*; *Bai et al., 2013*). Additional FoxO targets, such as peroxiredoxin 3 (*Chiribau et al., 2008*), were also upregulated in adults following developmental ΔOTC expression, but fell below our 3-fold cutoff. Similarly, targets that are negatively regulated by FoxO were found to be downregulated in response to developmental mitochondrial stress (Supplementary File 1).

## Persistent FoxO activation and lifespan are diet-dependent and regulated by mTORC1 and GCN-2 activity

To identify the biological mechanism promoting persistent FoxO activation, we first analyzed potential post-translational modifications of FoxO. It was previously reported that nicotinamide adenine dinucleotide (NAD+) precursors activate the UPR$^{mt}$ in worms, via the deacetylase *sir-2.1* (*Mouchiroud et al., 2013*). Knocking down *Sirt2* (the fly homologue of *sir-2.1* and *SIRT1*) during developmental ΔOTC expression, however, did not affect FoxO nuclear localization, and neither *Sir2* overexpression nor supplementation with the NAD precursor nicotinamide riboside were sufficient to induce nuclear translocation of FoxO (*Figure 5—figure supplement 1b–d*). *Drosophila* FoxO can also be regulated by HDAC4 (*Wang et al., 2011*), but knocking down this deacetylase during developmental ΔOTC expressionalso did not affect FoxO localization (*Figure 5—figure supplement 1e*).

We next asked whether lasting FoxO activation was the result of nutrient response signaling. FoxO activity is regulated by insulin signaling, in flies mediated by the *Drosophila* Insulin-like Peptide (*dilp)* family (*Kannan and Fridell, 2013*). During starvation, reduced availability of dILPs promotes FoxO nuclear translocation in the fat body, by reducing insulin receptor and protein kinase B activities. Upon refeeding, this signaling network is reset, promoting FoxO translocation to the cytoplasm. To test whether reduced insulin signaling activity was carried over from developmental stress into adulthood, we performed starvation and refeeding experiments on these flies. As shown in *Figure 5a*, FoxO translocates to the nucleus after developmental ΔOTC expression. After 9 hours of starvation (only water was offered to flies), the fat body of both stressed and control flies exhibit nuclear FoxO. 24 hr of refeeding resets this activation in control flies, but FoxO remained nuclear in flies that had experienced developmental stress. We were also unable to detect changes in transcription of *dilp2*, −3,−6 and −8 during/after ΔOTC expression. Further supporting the idea that the mitochondrial proteostatic stress affects FoxO independently of insulin signaling, switching flies to a diet of pure sucrose (5% in water) for 5 days did not abolish the persistent nuclear localization of FoxO caused by developmental ΔOTC expression (*Figure 5b*).

However, flies switched to an alternate diet containing ~5 x higher protein levels than control food showed a nearly complete loss of nuclear FoxO in the fat body after 5 days (*Figure 5b*). To test whether this high-protein diet affects FoxO localization through the main amino-acid-sensing pathway mTORC1, we repeated this experiment with supplementation of the mTORC1 inhibitor rapamycin at 200 μM. This prevented the loss of nuclear FoxO (*Figure 5b*), suggesting that mTORC1 activation can override the persistent nuclear translocation of FoxO produced by developmental ΔOTC expression. In support of this role for mTORC1, activating mTORC1 during developmental stress (by RNAi of its inhibitor TSC1) prevents lasting FoxO activation (*Figure 5c*, left), while reducing developmental survival in the same manner as FoxO deficiency (*Figure 5—figure supplement 2*). Furthermore, simultaneous over-expression of TSC1 and 2 to inhibit mTORC1 activity in adult flies is sufficient to promote nuclear translocation of FoxO (*Figure 5c*, right). mTORC1 activity has previously been reported to inhibit the FoxA ortholog Fork Head by preventing its nuclear localization (*Bülow et al., 2010*). This involves signaling through the protein kinase GCN-2, which has previously been implicated in the UPR$^{mt}$ in *C. elegans* (*Baker et al., 2012*). Indeed, overexpression of GCN-2 was sufficient to induce nuclear localization of FoxO, while knocking down GCN-2 during

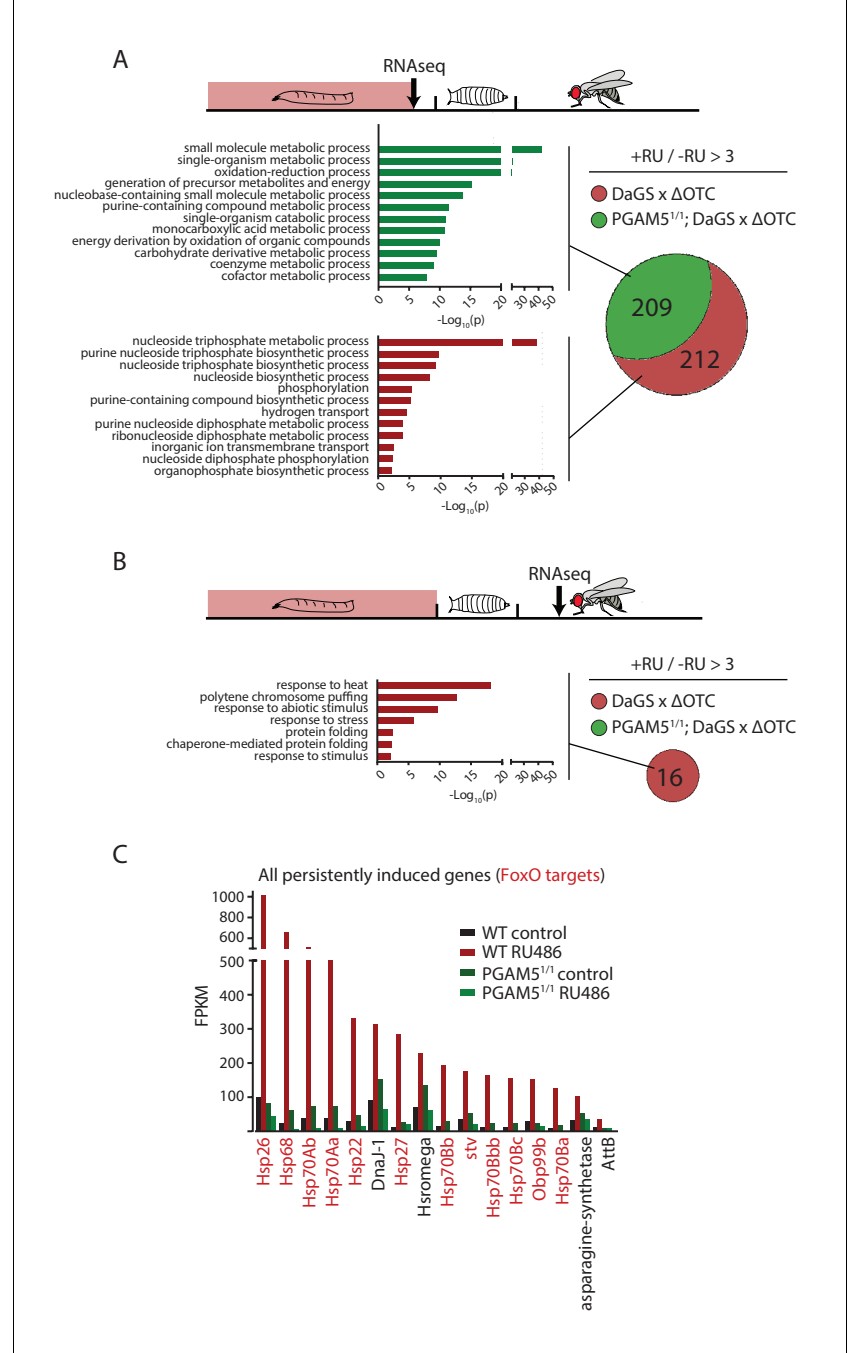

**Figure 4.** Lasting transcriptional activation of FoxO targets after developmental stress. (**A**) Venn diagram shows genes upregulated ≥3 x in fat bodies from third instar larvae undergoing mitochondrial proteostatic stress relative to untreated controls, where control FKMP ≥10. We separate genes whose upregulation depends on the identified stress signaling pathway (red) from those induced by other means (green), by comparing the transcriptional profile of PGAM5 null mutant larvae. Bar graphs show overrepresented Gene Ontology terms within each group of genes, showing mainly metabolic changes. Redundant GO terms were trimmed using REVIGO. (**B**) Venn diagram shows genes whose expression levels are persistently upregulated ≥3 x after developmental stress. All these genes were dependent on functional UPR$^{mt}$ signaling (i.e. not induced in PGAM5 nulls). Most of these genes correspond to heat-shock proteins. (**C**) Expression of the persistently upregulated genes is shown for all conditions, with FoxO target genes highlighted in red. Several of these have been shown to extend lifespan when overexpressed (see text). Full raw data found in Suppl. File 1. 200 µM RU486 was used.
DOI: https://doi.org/10.7554/eLife.26952.015

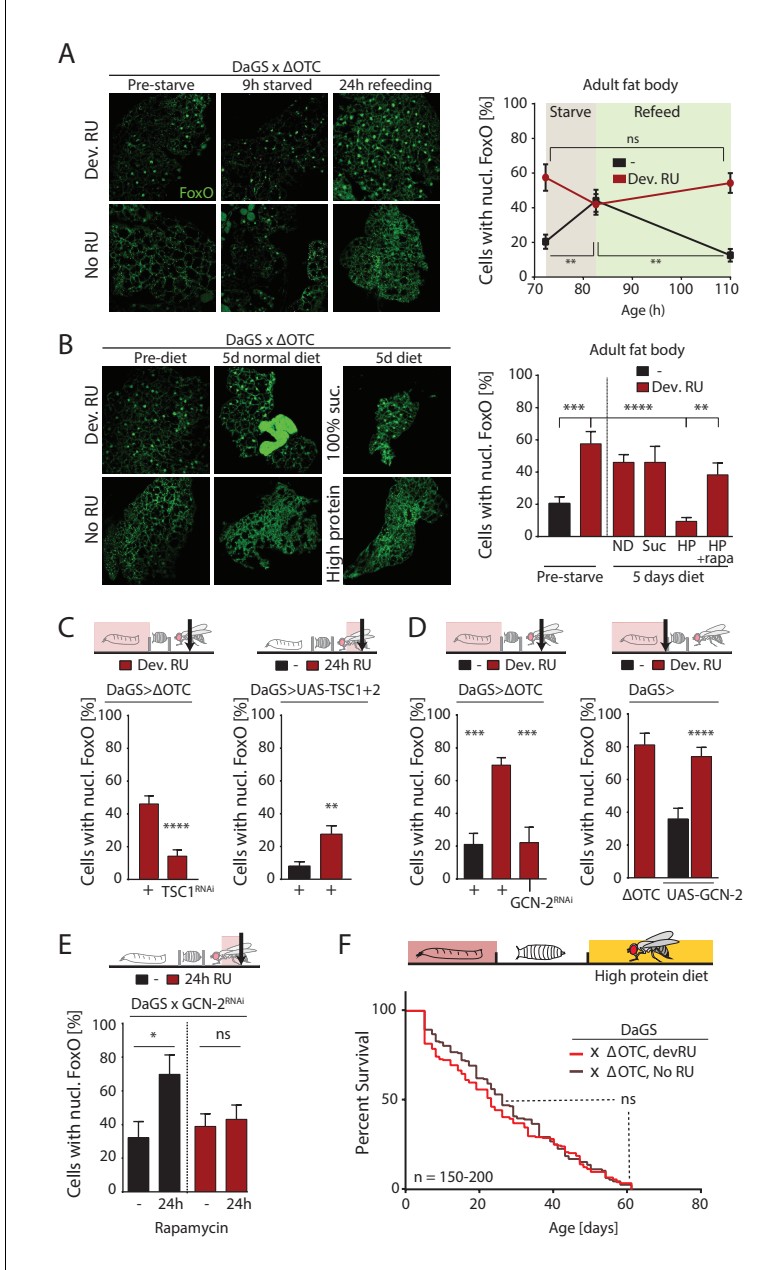

**Figure 5.** ΔOTC-mediated FoxO activity is regulated by diet, through mTORC1 and GCN-2. (**A**) Adult flies show nuclear localization of FoxO after developmental ΔOTC expression. After 9 hr of wet starvation, FoxO is nuclear in treated and untreated flies alike. 24 hr of refeeding returns FoxO to the cytoplasm in untreated flies, but does not reset the effect of developmental stress. p-values are (pre to starved) 0.517 and 0.031, (starved to refed) 0.723 and 0.001. (**B**) FoxO nuclear localization following developmental stress is reset by 5 days on a high-protein diet, but not a pure sucrose diet. Feeding rapamycin at 200 μM to inhibit mTORC1 with the high-protein diet blocks this effect. p-values are (±RU) 0.007, (+RU vs diets) 0.790, 0.915, <0.0001 and 0.338, (HPD ± rapa) 0.001. (**C**) Knocking down TSC1 during developmental ΔOTC expression blocks persistent activation of FoxO in adults (left). Conversely, overexpressing the mTORC1 inhibitors TSC1 and two in adult flies is sufficient to change FoxO localization absent mitochondrial stress (right). p-values are <0.0001 and 0.0027. (**D**) Knocking down GCN-2 during developmental ΔOTC expression is sufficient to block persistent FoxO activation (left). p-values are 0.0005 and 0.0006. Overexpressing the GCN-2 kinase in larvae produces FoxO nuclear localization similar to the effect of mitochondrial stress (right). p-values are (ΔOTC vs UAS-GCN2) 0.703 and (UAS-GCN2 ±RU)<0.0001. (**E**) Inhibiting mTOR with 200 μM rapamycin is sufficient to induce nuclear FoxO localization. This effect is blocked by RNAi of GCN-2, suggesting a downstream role for this kinase. p-values are 0.032 and 0.602. (**F**) Developmental ΔOTC
*Figure 5 continued on next page*

*Figure 5 continued*

expression does not extend lifespan when flies are fed a high-protein diet throughout adulthood, consistent with a requirement for persistent FoxO activity. All error bars are SEM from 2 + independent experiments. ANOVA with Tukey's (A-B) or dunnett's (D) post-hoc, or student's t-test (C and E); *p<0.05, **p<0.01, then each *=0.1 x. 200 µM RU486 used in all experiments.

DOI: https://doi.org/10.7554/eLife.26952.016

The following source data and figure supplements are available for figure 5:

**Source data 1.** ΔOTC-mediated FoxO activity is regulated by diet, through mTORC1 and GCN-2.
DOI: https://doi.org/10.7554/eLife.26952.019
**Figure supplement 1.** Lasting FoxO activation is not mediated by NAD-dependent histone deacetylases.
DOI: https://doi.org/10.7554/eLife.26952.017
**Figure supplement 2.** mTOR activation impairs survival through developmental stress.
DOI: https://doi.org/10.7554/eLife.26952.018

developmental ΔOTC expression abrogates the lasting effect on FoxO (*Figure 5d*). To test whether GCN-2 acts downstream of mTORC1 in this context, we knocked down GCN-2 in flies treated with rapamycin and observed a reduction in nuclear FoxO (*Figure 5e*).

Because FoxO activity is required for lifespan extension (*Figure 2d*) and a high-protein diet erases the persistent FoxO activation seen after developmental ΔOTC expression (*Figure 5b*), we hypothesized that adult diet would affect the longevity effects of mitochondrial proteostatic stres. To test this hypothesis, we repeated our lifespan experiment with developmental ΔOTC expression but switched flies to the high-protein diet after eclosion. The high-protein diet reduced the lifespan of the control group relative to our standard fly food, consistent with previous literature, but also completely abolished the lifespan extension normally induced by ΔOTC expression (*Figure 5f*). This supports our proposed pathway for lifespan extension, and suggests diet as an important regulator of beneficial UPR$^{mt}$ effects.

## Discussion

Our data identify a signaling pathway that responds to mitochondrial proteostatic stress through the phosphatase PGAM5, leading to activation of JNK and FoxO. This results in the FoxO- and Rel-mediated induction of immune, antioxidant and metabolic gene expression, as well as of genes encoding heat shock proteins. When activated during development, FoxO remains active throughout life, inducing a select group of genes that extend lifespan. Interestingly, this activity is subject to regulation by GCN-2/mTORC1-dependent nutrient sensing, providing a clue for how diet may interfere with lifespan extending stress signaling mechanisms (*Figure 6*).

Our genetic studies suggest a pathway of UPR$^{mt}$ activation from the mitochondrial membrane protein PGAM5, through ASK1 and JNK, to the FoxO transcription factor. FoxO increases the expression of Relish, and thereby induces antimicrobial peptide expression. Persistent activation of FoxO also leads to lasting upregulation of chaperones, which improves proteostasis and extends lifespan. Meanwhile, sensing of amino acids can activate mTORC1 and GCN-2, which negate the persistent activation of FoxO and block lifespan extension.

PGAM5 has not been fully characterized but plays a role in regulating cell death pathways in flies and cultured cells (*Ishida et al., 2012*; *Wang et al., 2012*; *Zhuang et al., 2013*). It has been reported to localize to either the inner (*Lo and Hannink, 2008*) or outer mitochondrial membrane, where it is cleaved upon loss of membrane potential (*Sekine et al., 2012*). While a member of the PGAM family based on sequence, it displays phosphatase rather than mutase activity in culture (*Takeda et al., 2009*). PGAM5 also interacts with the mitophagy factor PINK1, and loss of *PGAM5* rescues the muscle degeneration phenotype of *dPINK1* mutant flies (*Imai et al., 2010*). Loss of PGAM5 inhibits mitophagy in vitro, and leads to Parkinson's-like symptoms in mice (*Lu et al., 2014*). Regulation of mitophagy by PGAM5 may depend on its interaction with the PARL protease, which is normally responsible for cleaving PINK1 to prevent mitophagy of healthy mitochondria (*Sekine et al., 2012*). Indeed, induction of the UPR$^{mt}$ in HeLa cells has been reported to trigger PINK1 accumulation, Parkin recruitment and mitophagy (*Jin and Youle, 2013*). In our experiments, PGAM5 does not seem to regulate longevity directly by stimulating apoptosis, however, suggesting

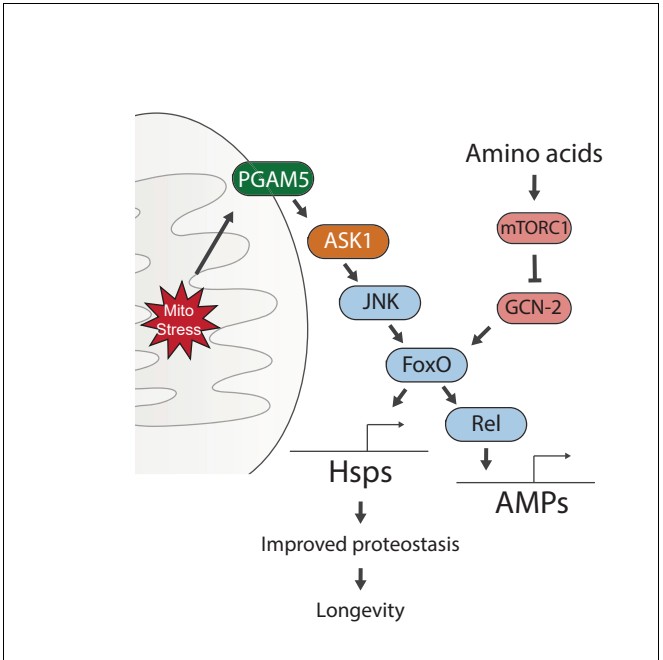

**Figure 6.** Proposed model for UPR[mt]-mediated longevity.
DOI: https://doi.org/10.7554/eLife.26952.020

a more complicated biological role for this phosphatase. It is tempting to speculate that it may act as a rheostat to dictate whether mitochondrial stress results in UPR[mt] activation, mitophagy, or cell death. Further exploration of the intra-mitochondrial signals that activate PGAM5, as well as its immediate downstream partners, should be interesting topics for future studies of mitochondrial stress.

The importance of developmental activation of the UPR[mt] for lifespan extension was previously observed in studies exploring this mitochondrial stress response and the response to ETC dysfunction in *C. elegans* and *Drosophila* (*Copeland et al., 2009*; *Dillin et al., 2002*; *Durieux et al., 2011*; *Rea et al., 2007*). Our findings confirm this and identify persistent FoxO activation and FoxO-induced gene expression as associated with the lasting benefits of developmental mitochondrial stress in the adult organism. Two recent papers have identified a role for the UPR[mt] in regulating histone methylation and chromatin remodeling (*Merkwirth et al., 2016*; *Tian et al., 2016*). The studies show that demethylase activity during developmental UPR[mt] is required for lifespan extension, and that *atfs-1* acts synergistically with these epigenetic changes. It will be interesting to explore the interplay between elevated FoxO activity and demethylase-regulated chromatin accessibility in future studies.

The hypothesis that a combination of signaling and epigenetic changes are required for UPR[mt] longevity could help explain recent reports that commonly used UPR[mt] reporters in *C. elegans* correlate poorly with longevity (*Bennett et al., 2014*; *Ren et al., 2015*): the UPR[mt] likely involves several pathways, which can also be activated by other stimuli. However, a crucial combination of signaling and epigenetic changes could be induced by specific types of mitochondrial stress. This would prompt caution about using a single gene reporter as a measure of 'UPR[mt] activation' and suggest using multiple assays to determine its role in each biological process. To our knowledge, this is the first report of an organism other than *C. elegans* showing improved longevity from UPR[mt] activation that does not involve direct ETC disruption. This lends support to the UPR[mt] as a conserved longevity mechanism, which overlaps with but is not fully explained by ETC function.

Our work also provides further support for the observation that the UPR[mt] triggers the innate immune system (*Pellegrino et al., 2014*) and shows that this feature is conserved outside of *C. elegans*. As in worms, this activation provides acute resistance to infection in flies, but we find that immune function is not responsible for the effects of the UPR[mt] on longevity. The activation of

antimicrobial gene expression in response to FoxO activation is not surprising, as it has previously been reported that the FoxO transcription factor can induce the expression of antimicrobial peptides independently of classical innate immune pathways in response to starvation (*Becker et al., 2010*). Moreover, we have shown previously that FoxO activation in the larval *Drosophila* fat body can induce *Relish*/*NF-κB* signaling by stimulating *Rel* expression (*Karpac et al., 2011*). Our data indicate that mitochondrial proteostatic stress activates this FoxO/Relish cassette, but that the FoxO-dependent upregulation of chaperones in adult flies is likely responsible for lifespan extension.

Our results are also consistent with reports that the UPR^mt can be activated by rapamycin (*Houtkooper et al., 2013*) and identify high-protein diet as a critical intervention that activates mTORC1 and overrides the positive effects of developmental mitochondrial stress. It was also previously reported that NAD precursors and *sirt-2.1* activate the UPR^mt and extend lifespan in a *daf-16/FOXO*-dependent manner (*Mouchiroud et al., 2013*). Our experiments did not suggest a role for NAD precursors or *Sirt2* in regulating the *Drosophila* UPR^mt, but it remains possible that exploring different sirtuins or dietary conditions could reveal such a role. This is especially true in light of the observed effect of diet on UPR^mt activity and lifespan extension. The regulation of FoxO by amino acid sensing/mTORC1 rather than insulin signaling is surprising, but is supported by a previous observation that mTORC1 activity regulates longevity in *C. elegans* through activation of *daf-16/FOXO* (*Robida-Stubbs et al., 2012*).

These findings thus reveal a critical vulnerability of developmental UPR^mt-mediated physiological changes that promote longevity: they can be erased by a high-protein diet. If the observed mTORC1-induced signaling interactions are conserved in vertebrates, this has important implications for the development of interventions that aim to engage the UPR^mt to increase metabolic health and extend health- or lifespan.

## Materials and methods

### Drosophila stocks

The following lines were obtained from the Bloomington *Drosophila* Stock Center: $w^{1118}$, $y^1 w^1$, Da::GS, tub::GS, *PGAM5* RNAi (#34744), *ASK1* RNAi (#32646), $foxo^{21}$ (*Jünger et al., 2003*), $foxo^{\Delta 94}$ (*Slack et al., 2011*), $bsk^{DN}$ (#6409).

The following lines were obtained from the Vienna *Drosophila* RNAi Center: *Hrd1* RNAi (#6870), *PGAM5* RNAi (#51657), *ASK1* RNAi (#110228), *TSC1* RNAi (#110811), *bsk* RNAi (#34138), *foxo* RNAi (#30556).

The following lines were gifts from other labs: UAS-ΔOTC (Martins lab, *Pimenta de Castro et al., 2012*), $PGAM5^1$ (Ichijo lab, *Imai et al., 2010*), *jun* and *fos* RNAi (Bohmann lab, *Hyun et al., 2006*)), UAS-*GCN-2* and *GCN-2* RNAi (Leopold lab, *Bjordal et al., 2014*), UAS-TSC1 and TSC2 (Tatar lab, *Hwangbo et al., 2004*), *ND75* and *Rel* RNAi (Perrimon lab, *Agaisse et al., 2003*; *Owusu-Ansah et al., 2013*)

### Fly husbandry and demographics

Standard fly food was prepared with the following recipe: 1 l distilled water, 13 g agar, 22 g molasses, 65 g malt extract, 18 g brewer's yeast, 80 g corn flour, 10 g soy flour, 6.2 ml propionic acid, 2 g methyl-p-benzoate in 7.3 ml of EtOH. High-protein food was prepared with the following recipe: 1 l distilled water, 10 g agar, 80 g brewer's yeast, 20 g yeast extract, 20 g peptone, 51 g sucrose, 6.2 ml propionic acid, 2 g methyl-p-benzoate in 7.3 ml of EtOH (*Musselman et al., 2011*). 100% sucrose diet is 50 g/l sucrose in water. For GeneSwitch experiments, 86 mg RU486 was additionally dissolved in the EtOH for 200 μM final concentration. For antibiotic experiments, 50 mg of the following antibiotics were added during food preparation: ampicillin, tetracycline, erythromycin, kanamycin. For axenic experiments, fly food bottles were autoclaved at 121°C for 30 min, and 1 ml EtOH ±5 mg RU486 added after cooling, previously determined to yield a final concentration of 200 μM (*Biteau et al., 2010*). Eggs laid over a 24 hr period were collected and sterilized for 3 min in 2.7% sodium hypochlorite, then washed twice with sterile, distilled water for 1 min and transferred to sterile bottles in a laminar flow hood. Eclosed flies were kept on sterilized food in the flow hood. Flies were maintained at 25°C and 65% humidity, on a 12 hr light/dark cycle, unless otherwise indicated.

For lifespan experiments, up to 30 flies per vial were flipped thrice weekly, with dead flies counted visually.

## Immunostaining and microscopy

Fat bodies were dissected from adult females or from third instar larvae pre-wandering. Tissue was fixed at room temperature for 45 min in 100 mM glutamic acid, 25 mM KCl, 20 mM MgSO4, 4 mM Sodium Phosphate, 1 mM MgCl2, 4% formaldehyde. Washes were done in PBS with 0.5% BSA, 0.1% Triton X-100 at 4°C. Primary incubation was done overnight 4°C using a rabbit N-terminal FoxO antibody (RRID:AB_2569227, gifted from O. Puig) at 1:500. Secondary antibodies were from Jackson Immunoresearch. TUNEL staining was performed using In Situ Cell Death Detection Kit (Roche). DNA was stained using Hoechst, and tissue mounted on slides using Mowiol mounting medium (Sigma). Imaging was done on a Zeiss LSM700 confocal microscope at 40x magnification.

## NAD/metabolism assays

All assays were done on samples of five whole flies/larvae. NAD and NADH levels were quantified from whole flies using the NAD/NADH Quantitation Colorimetric kit from BioVision. Glucose and glycogen were quantified using the Glucose (HK) Assay kit from Sigma. Lipids were quantified with the Triglyceride LiquiColor kit from Stanbio. Protein levels were quantified with the BCA assay.

## Infection

*P. entomophila* and *P. aeruginosa* were cultured in LB medium for 16 hr at 37°C and 10 hr at 30°C, respectively. For oral infections, flies were starved for 2 hr before treatment. 6 mL culture was pelleted at 5000 g for 10 min, then resuspended in 500 uL 5% sucrose and added to a vial containing Whatman filter paper. 20 flies per vial were maintained at 29°C, and deaths tracked daily or twice daily. 100 uL 5% sucrose was added to the vials each day. For humeral infections, a tungsten needle was dipped in 30x concentrated culture and flies were pricked in the periphery of the abdomen. They were then transferred to normal food vials. Negative controls were fed/pricked with 5% sucrose alone.

## RNA extraction and quantitative RT-PCR analysis

Muscle tissue was isolated by removal of head, abdomen, wings, legs and intestine from fly thoraces. Fat bodies were manually isolated from fixed abdomens/larvae. 5–8 flies were used per sample, and total RNA was extracted from using Trizol (Life Sciences) according to manufacturer's protocol. cDNA was synthesized using Superscript III (Life Sciences). Results represent four biological samples, each with triplicate technical repeats. Real-time PCR was performed using SsoAdvanced Universal SYBR Green Supermix (Bio-Rad) on a Bio-Rad Real-Time CFX96 system. Expression levels were calculated as ΔΔCt normalized to *RP49*.

## RNA sequencing

Muscle or fat body tissue was isolated and RNA extracted as described above, using 15 flies per sample. TruSeq RNA Library Prep Kit (Illumina) was used to prepare libraries, and sequencing was performed on an Illumina MiSeq system. Raw data were analyzed with the Tuxedo suite (RRID:SCR_013194) and reads were mapped to Drosophila genome release 5.2. Expression was recorded as FPKM: fragments per kilo-base per million reads.

## Statistics and bioinformatics

Significance in two-condition experiments was evaluated by student's t-test. Multiple condition experiments were evaluated by one-way ANOVA, with Dunnett's post-hoc comparing to induced sample or Tukey's post-hoc when multiple conditions are compared. Gene Ontology analysis of RNAseq experiments was done at flymine.org, using REVIGO (RRID:SCR_005825) to trim redundant terms (Allowed similarity: Medium).

## CAFE assay

Feeding rates were quantified by the CAFE assay (*Ja et al., 2007*), measuring consumption of a 5% w/v sucrose solution accessible from a capillary tube. Each vial contained 10 flies per vial, and

evaporation was measured with an empty vial. Consumption was measured at 8–12 hr intervals and normalized to uL/fly/hr.

## 16S sequencing

Flies were washed in 70% EtOH for 60 s to kill external bacteria, and intestines dissected out in sterile PBS. Ten guts were collected per sample. Commensal genomic DNA was extracted using Ultra-Clean Microbial DNA Isolation Kit (MO BIO). This DNA was used as template for limited cycle PCR with primers targeting V3/V4 regions (Forward 5'-CCTACGGGNGGCWGCAG-3' and Reverse 5'-GACTACHVGGGTATCTAATCC −3') to generate a 16S metagenomic sequencing library. The following reaction conditions were used: 94°C for 5 min, followed by 30 cycles of 94°C for 1 min, 48°C for 2 min, and 72°C for 2 min, and a final extension at 72°C for 5 min. Illumina Miseq paired-end (2 × 300 bp) sequencing was performed and Miseq Reporter Software was used for primary analysis and classification based on Greengenes database (RRID:SCR_002830).

## Commensal quantification

Intestines were dissected out as described above, pooled in groups of 10 and crushed with a motorized pestle. These samples were diluted 1000x and plated on non-selective LB Amp plates. Sterile PBS was used as a negative control. Colonies were counted after 60 hr at 30°C.

## Primers used

*RP49* F: 5'-TCCTACCAGCTTCAAGATGAC-3'
*RP49* R: 5'-CAGGTTGTGCACCAGGAACT-3'
*Metchnikowin* F: 5'-AATCAATTCCCGCCACCGAG-3'
*Metchnikowin* R: 5'-GACCCGGTCTTGGTTGGTTA-3'
*Drosomycin* F: 5'-CGTGAGAACCTTTTCCAATATGAT-3'
*Drosomycin* R: 5'-TCCCAGGACCACCAGCAT-3'
*Turandot A* F: 5'-GCACCCAGGAACTACTTGACATCT-3'
*Turandot* A R: 5'-GACCTCCCTGAATCGGAACTC-3'
*Relish* F: 5'-ACAGCCCACATTCCCATCAG-3'
*Relish* R: 5'-GAGCCGCACCTGGTTCAA-3'
*Hsp60* F: 5'-GACCAGATCGAGGACACCAC-3'
*Hsp60* R: 5'- GCCGAGTTTCTGATCCTCGTTG-3'
*Hsc70-5* F: 5'-CTGCGTTACAAGTCCGGTGA −3'
*Hsc70-5* R: 5'- GCAGCACATTAAGACCAGCG-3'
*Hsp10* F: 5'-CCCGCATCTAGCGAGAATAG −3'
*Hsp10* R: 5'-CTCCTTTCGTCTTGGTCAGC −3'
*ClpX* F: 5'-AAAATGCTCGAAGGCACAGT −3'
*ClpX* R: 5'-TTGAGACGACGTGCGATAAG −3'

## Acknowledgements

This work was supported by the National Institute on Aging (NIH R01 AG028127 and R01 AG050104), the American Federation for Aging Research (Breakthroughs in Gerontology award to HJ), and by an Alfred Benzon fellowship to MBJ. We thank Drs. Oscar Puig, Hidenori Ichijo, Dirk Bohmann, Pierre Leopold, Samantha H Loh, Norbert Perrimon, Marc Tatar and the Bloomington and Vienna stock centers for antibodies and flies. We thank Suzy Jackson for assistance with metabolite experiments. We further thank Olivia Murillo, Maurice Brady, Margot Devincenzi and Meredith Nix for fly maintenance.

## Additional information

### Funding

| Funder | Grant reference number | Author |
|---|---|---|
| National Institute on Aging | R01 AG028127 | Yanyan Qi<br>Rebeccah Riley<br>Heinrich Jasper |
| American Federation for Aging Research | Breakthroughs in Gerontology Award | Heinrich Jasper |
| Alfred Benzon Foundation | Postdoctoral Fellowship | Martin Borch Jensen |
| National Institute on Aging | R01 AG050104 | Yanyan Qi<br>Rebeccah Riley<br>Heinrich Jasper |

The funders had no role in study design, data collection and interpretation, or the decision to submit the work for publication.

### Author contributions

Martin Borch Jensen, Conceptualization, Data curation, Formal analysis, Supervision, Funding acquisition, Investigation, Methodology, Writing—original draft, Project administration, Writing—review and editing; Yanyan Qi, Investigation, RNA and 16S sequencing experiments; Rebeccah Riley, Investigation, RNA sequencing experiments; Liya Rabkina, Investigation, Metabolite experiments; Heinrich Jasper, Conceptualization, Data curation, Formal analysis, Supervision, Funding acquisition, Methodology, Project administration, Writing—review and editing

### Author ORCIDs

Martin Borch Jensen (iD) http://orcid.org/0000-0002-8875-0345
Heinrich Jasper (iD) http://orcid.org/0000-0002-6014-4343

### Decision letter and Author response

Decision letter https://doi.org/10.7554/eLife.26952.024
Author response https://doi.org/10.7554/eLife.26952.025

## Additional files

### Supplementary files

• Supplementary file 1. Raw data for RNA sequencing experiments in *Figure 1 and 4*.
DOI: https://doi.org/10.7554/eLife.26952.021

• Supplementary file 2. Life tables for all longevity experiments.
DOI: https://doi.org/10.7554/eLife.26952.022

• Transparent reporting form
DOI: https://doi.org/10.7554/eLife.26952.023

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
