## [Decision Letter]

Thank you for submitting your article "PGAM5 promotes lasting FoxO activation after developmental mitochondrial stress and extends lifespan in *Drosophila*" for consideration by *eLife*. Your article has been favorably evaluated by Jonathan Cooper (Senior Editor) and three reviewers, one of whom, Utpal Banerjee, is a member of our Board of Reviewing Editors. The following individual involved in review of your submission has agreed to reveal their identity: Edward Owusu-Ansah (Reviewer #2).

The reviewers have discussed the reviews with one another and appreciate the importance of the findings and the overall quality of the data. The summary of these reviews and discussions are summarised below by the Reviewing Editor.

A set of specific issues raised by the reviewers that we believe can be addressed by new experiments and/or rewriting the manuscript are summarized below as "other concerns". However, there is one major issue that needs special attention. During the post-review discussions, the reviewers unanimously agree that the data lack direct evidence for a sustained FOXO activation (which, as the title suggests, is central to the conclusions derived from this work).

Summary:

In this paper Jensen et al. investigate how mitochondrial dysfunction during *Drosophila* development leads to a stress response that has lasting effects on the flies, most notably on extension of lifespan. The authors characterize the steps that signal mitochondrial dysfunction to modulate nuclear gene expression, using an immune response gene as an output. In adult flies that were subjected to mitochondrial stress during development, a subset of stress response genes are expressed (even though the original stimulus is removed); it is proposed that these genes (all FoxO targets) mediate lifespan extension. The lasting effects of developmental mitochondrial dysfunction can be reversed through exposure to a high protein diet during adulthood.

Essential revisions:

(major concerns):

The key weakness regards the study's (implied) inference on the role of lasting FoxO activation as the longevity mechanism. Epistasis analysis is needed to test this idea.

The important required study is to conduct mtUPR (in any way) in FoxO-null epistasis. Alternatively, the authors could test the effect of co-expression of the OTC construct and RNAi to FoxO, using the gene-switch system. This approach allows the genotype without RU to serve as its own co-isogenic control. One needs to measure the longevity of the OTC expression alone, using geneswitch again, to confirm there is a longevity benefit in that period. And, one should do the same by driving the FoxO RNAi with the same geneswitch (which should do very little to the survival).

Essential revisions:

(other concerns):

1) Are phenotypes due to a UPR^mt^ stress response per se? How much is due to a more general response caused by mitochondrial dysfunction. This is unclear because of the nature of mitochondrial stressors. ND75 knockdown will cause mitochondrial dysfunction (possibly affecting ATP, Ca^2+^, ROS, NADH, NAD+ etc.). The ΔOTC mutant in flies also severely impacts mitochondrial function (ATP, oxygen consumption etc.). Therefore, it is not clear that a UPR^mt^ or some other type of mitochondrial stress response is being studied. Presumably studies in worms, where chaperone up-regulation has clear linkage to UPR^mt^ provides the inspiration, but this is not yet in flies. For example, ΔOTC causes ETC dysfunction in flies (Pimenta de Castro et al. 2012). If this is not true for this system, then some functional data are required, or the description of the model should be modified.

2) Is the PGAM5/ASK/JNK signaling pathway a bona fide UPR^mt^ pathway (analogous to the ATFS-1)? If a chaperone important in the UPR^mt^ response affecting proteostasis directly was chosen as a marker rather than a gene involved in the immune response, then the relation of this new signaling pathway to UPR^mt^ could be better assessed. Either a more specific UPR^mt^ output marker should be used, or discussion of UPR^mt^ should be reserved for the Discussion.

3) In the same vein, it is unclear whether the canonical markers of the UPR^mt^, such as Hsp60, Hsp10, mortalin (mt-Hsp70) and the ClpP protease are induced under the conditions used to activate the UPR^mt^ in this study. The authors should comment on this, and provide a possible explanation for this apparent disparity.

4) The exact dosages of RU456 used in the gene-switch experiments should be listed in the figure legends.

5) The link between UPR^mt^ and longevity is not yet settled, and while this work provides important advancements, the authors will do well to tone down the enthusiasm with which they link expression of the OTC construct with UPR^mt^ induction. OTC induces other genes as well (such as antimicrobial peptides). Therefore several bold statements such as the subheading "UPR^mt^-mediated longevity is not caused by improved adult immune function, or by changes to the microbiome" and "Developmental UPR^mt^ affects the metabolic state of adult flies and leads to persistent FoxO" or (based on point 1 above), "Of the two methods initially used to induce the UPR^mt^, we opted to use ΔOTC for further experiments to avoid potential secondary effects of inducing ETC dysfunction" should be modified to reflect the fact that the results observed are due to OTC expression and not necessarily all due to the UPR^mt^.

6) ADaGSxOTC itself yields less than 50% eclosion. Survivors of this cohort may live longer because of frailty selection: weak larvae that would produce shorter-lived adults do not eclose. Some rescue-type data (Figure 2) argues against this potential confound, but the evidence is thin.

7) UPR^mt^ induced in adults activates FoxO, but such cohorts are not long-lived when RU is constantly applied. That FoxO is only transiently activated when RU is given for just one week is not a satisfactory explanation.

8) The data with HPD is unclear. Presumably, HPD resets both this TF and the lifespan, and this is used to infer causality between the induced longevity and FoxO activation. But in the sole survival experiment to this point (Figure 5), the shape of the plots is a concern: they are too linear, suggesting that age-independent mortality is the overriding cause of death. This can mask any potential impact on age-dependent mortality; and there are no data to rule this out (or in). Perhaps the HPD is toxic, and all flies die for reasons besides aging. There are no other data to address the relevance of activated FoxO (by any of the interesting, observed mechanisms) as relevant to the larval UPR^mt^ impact on adult longevity.

---

## [Author Response]

Essential revisions:(major concerns):The key weakness regards the study's (implied) inference on the role of lasting FoxO activation as the longevity mechanism. Epistasis analysis is needed to test this idea.The important required study is to conduct mtUPR (in any way) in FoxO-null epistasis. Alternatively, the authors could test the effect of co-expression of the OTC construct and RNAi to FoxO, using the gene-switch system. This approach allows the genotype without RU to serve as its own co-isogenic control. One needs to measure the longevity of the OTC expression alone, using geneswitch again, to confirm there is a longevity benefit in that period. And, one should do the same by driving the FoxO RNAi with the same geneswitch (which should do very little to the survival).

We agree that this is an important experiment. Per the reviewer’s suggestions, we have now carried out duplicate lifespan experiments where DaGS was used to drive ΔOTC expression in a FoxO null background (the foxo^Δ94/21^ transheterozygotes already described in the manuscript). In parallel, we replicated the DaGS x ΔOTC lifespan experiment already included in the paper. In support of our model, developmental mitochondrial stress leads to a significant shortening of lifespan in FoxO null flies, in sharp contrast to the lifespan extension observed in normal flies. The FoxO null data is presented in the revised Figure 2, alongside statistics for the positive control in 2G. The graph for the positive control is shown in Figure 2—figure supplement 2. Combined with our observations of drastically reduced survival through developmental mitochondrial stress in FoxO nulls (now in Figure 2—figure supplement 1), this solidifies FoxO activity as a necessary factor for the lifespan extension elicited by developmental mitochondrial stress.

Essential revisions:(other concerns):1) Are phenotypes due to a UPR^mt^ stress response per se? How much is due to a more general response caused by mitochondrial dysfunction. This is unclear because of the nature of mitochondrial stressors. ND75 knockdown will cause mitochondrial dysfunction (possibly affecting ATP, Ca^2+^, ROS, NADH, NAD+ etc.). The ΔOTC mutant in flies also severely impacts mitochondrial function (ATP, oxygen consumption etc.). Therefore, it is not clear that a UPR^mt^ or some other type of mitochondrial stress response is being studied. Presumably studies in worms, where chaperone up-regulation has clear linkage to UPR^mt^ provides the inspiration, but this is not yet in flies. For example, ΔOTC causes ETC dysfunction in flies (Pimenta de Castro et al. 2012). If this is not true for this system, then some functional data are required, or the description of the model should be modified.

We agree that distinguishing general mitochondrial stress from a more specific UPR^mt^ is difficult in any organism. In our original manuscript, we decided to follow the nomenclature of previous studies (Pimenta de Castro et al. 2012, Owusu-Ansah et al. 2013). However, to our knowledge (and as suggested by the reviewers), all methods of inducing the UPR^mt^ in any organism leads to mitochondrial dysfunction of some sort. It does not yet seem clear whether and which such dysfunctions should be classified as causative or part of the UPR^mt^. For example, some parts of the UPR^mt^ have been characterized as ROS-dependent. We originally chose to define the UPR^mt^as any response occurring in two distinct types of mitochondrial proteostatic stress, but we agree with the reviewers’ point that the *Drosophila* UPR^mt^ remains imperfectly described. We have thus rewritten several parts of the manuscript to refer to ‘mitochondrial stress’ rather than UPR^mt^.

2) Is the PGAM5/ASK/JNK signaling pathway a bona fide UPR^mt^ pathway (analogous to the ATFS-1)? If a chaperone important in the UPR^mt^ response affecting proteostasis directly was chosen as a marker rather than a gene involved in the immune response, then the relation of this new signaling pathway to UPR^mt^ could be better assessed. Either a more specific UPR^mt^ output marker should be used, or discussion of UPR^mt^ should be reserved for the Discussion.3) In the same vein, it is unclear whether the canonical markers of the UPR^mt^, such as Hsp60, Hsp10, mortalin (mt-Hsp70) and the ClpP protease are induced under the conditions used to activate the UPR^mt^ in this study. The authors should comment on this, and provide a possible explanation for this apparent disparity.

Although the fluorescent reporters available for *hsp-6* and *hsp-60* in *C. elegans* respond strongly to induction of the UPR^mt^, *Drosophila* studies (Pimenta de Castro et al. 2012, Owusu-Ansah et al. 2013) have shown only moderate (~1.5-fold) induction of these genes. For this reason, the markers mentioned by reviewers were not included in the set of 3-fold upregulated genes in Figure 1, and we chose *metchnikowin* as our marker for subsequent screens due to its higher dynamic range of induction.

To address the pertinent comments by the reviewers, we have added a supplement to Figure 1 (Figure 1—figure supplement 1) showing that expression of ΔOTC using the regime reported in Owusu-Ansah et al. leads to a similar upregulation of hsp60, hsc70-5 and hsp10. We further show that this induction is blocked in the absence of PGAM5, bsk or FoxO.

4) The exact dosages of RU456 used in the gene-switch experiments should be listed in the figure legends.

200 μm RU486 was used throughout, and this has now been added to figure legends.

5) The link between UPR^mt^ and longevity is not yet settled, and while this work provides important advancements, the authors will do well to tone down the enthusiasm with which they link expression of the OTC construct with UPR^mt^ induction. OTC induces other genes as well (such as antimicrobial peptides). Therefore several bold statements such as the subheading "UPR^mt^-mediated longevity is not caused by improved adult immune function, or by changes to the microbiome" and "Developmental UPR^mt^ affects the metabolic state of adult flies and leads to persistent FoxO" or (based on point 1 above), "Of the two methods initially used to induce the UPR^mt^, we opted to use ΔOTC for further experiments to avoid potential secondary effects of inducing ETC dysfunction" should be modified to reflect the fact that the results observed are due to OTC expression and not necessarily all due to the UPR^mt^.

As noted in our response to comment #1, we have rewritten parts of the manuscript to refer to mitochondrial stress rather than UPR^mt^. In response to the comment that “OTC induces other genes as well (such as antimicrobial peptides).”, we note that Pellegrino et al. 2014 showed that AMPs are induced by ATFS-1 activation even absent mitochondrial stress, and were thus described as part of the core UPR^mt^ in *C. elegans*.

6) ADaGSxOTC itself yields less than 50% eclosion. Survivors of this cohort may live longer because of frailty selection: weak larvae that would produce shorter-lived adults do not eclose. Some rescue-type data (Figure 2) argues against this potential confound, but the evidence is thin.

This is an important concern, and indeed activation of the UPR^mt^ by constitutively active ATFS-1 in *C. elegans* also leads to reduced survival through development (Cole Haynes, personal communication). However, our new FoxO null lifespan data shows that FoxO activity not only correlates with, but is required for lifespan extension. It is pertinent to this specific concern that while FoxO null mutants show strongly reduced survival through developmental ΔOTC expression (Figure 2—figure supplement 1), even the survivors showed reduced adult lifespan (Figure 2). The same is true for PGAM5 nulls (Figure 2 and Figure 2—figure supplement 1).

7) UPR^mt^ induced in adults activates FoxO, but such cohorts are not long-lived when RU is constantly applied. That FoxO is only transiently activated when RU is given for just one week is not a satisfactory explanation.

We apologize for not making this clear in the original manuscript, as indeed the transience of FoxO activation by mitochondrial stress does not explain why continuous mitochondrial stress in adults does not extend lifespan. Our interpretation is that the negative effects of chronically impairing mitochondrial function through expression of ΔOTC has detrimental effects that overshadow the improved proteostasis conferred by activating the FoxO pathway. This is consistent with observations by Pimenta-Castro et al. 2012, who show impaired mitochondrial function and survival with chronic adult ΔOTC expression. In the case of developmentally expressed ΔOTC, on the other hand, the mitochondrial stress is eliminated in adults while FoxO activation is retained chronically.

8) The data with HPD is unclear. Presumably, HPD resets both this TF and the lifespan, and this is used to infer causality between the induced longevity and FoxO activation. But in the sole survival experiment to this point (Figure 5), the shape of the plots is a concern: they are too linear, suggesting that age-independent mortality is the overriding cause of death. This can mask any potential impact on age-dependent mortality; and there are no data to rule this out (or in). Perhaps the HPD is toxic, and all flies die for reasons besides aging. There are no other data to address the relevance of activated FoxO (by any of the interesting, observed mechanisms) as relevant to the larval UPR^mt^ impact on adult longevity.

As mentioned above, we have now obtained additional evidence that FoxO activity is required for lifespan extension by developmental ΔOTC expression (new lifespan data for null mutants). We have edited the manuscript to merely note that the lack of extension on HPD is consistent with our demonstration that HPD blocks FoxO activation and with the fact that FoxO is required for lifespan extension by developmental mitochondrial stress.